# Antimicrobial resistant bacteria isolated from Buruli ulcer lesions in Ghana

Nancy Ackam[1,2], Charity Wiafe Akenten[1,2], Abigail Opoku Boadi[1], Abigail Agbanyo[1], Kabiru Mohammed Abass[3], George Amofa[4], Elizabeth Ofori[5], Joseph Azabire[6], Augustina Sylverken[1,2], Kwasi Obiri-Danso[2], Mark Wansbrough-Jones[7], Thorsten Thye[8], Denise Dekker[8], Yaw Ampem Amoako [1,9,10]*, Richard Odame Phillips[1,8,9,10]

1 Kumasi Centre for Collaborative Research in Tropical Medicine (KCCR), Kumasi, Ghana, 2 Department of Theoretical and Applied Biology, Kwame Nkrumah University of Science and Technology, Kumasi, Ghana, 3 Agogo Presbyterian Hospital, Agogo, Ghana, 4 Dunkwa Government Hospital, Dunkwa, Ghana, 5 Tepa Government Hospital, Tepa, Ghana, 6 Wassa Akropong District Hospital, Wassa Akropong, Ghana, 7 Institute of Infection and Immunity, St George's University of London, London, United Kingdom, 8 Bernhard Nocht Institute for Tropical Medicine (BNITM), Hamburg, Germany, 9 School of Medical Sciences, Kwame Nkrumah University of Science and Technology, Kumasi, Ghana, 10 Komfo Anokye Teaching Hospital, Kumasi, Ghana

* yamoako2002@yahoo.co.uk

## Abstract

### Background

We previously showed that the presence of secondary bacteria influences clinical outcome in Buruli ulcer (BU) patients. Despite this, there is limited data on the antimicrobial resistance of these bacterial isolates within BU lesions. To gain understanding of antimicrobial resistance in BU, we longitudinally profiled antimicrobial resistance in frequently isolated bacterial organisms from these lesions.

### Methodology/Principal findings

Between August 2021 and June 2024, we assessed the antimicrobial resistance of pathogenic bacterial isolates within lesions of laboratory confirmed BU patients in Ghana. Wound swabs were collected longitudinally. The bacteria were identified and their antibiotic susceptibility tested using the VITEK 2 compact. Of the 166 bacterial isolates, eight bacterial species were identified comprising 56.9% Gram negative bacilli and 43.1% Gram positive cocci. We found the presence of pathogenic bacteria with varying levels of resistance to commonly used antibiotics in BU lesions before, during and after BU-specific antibiotic treatment. At baseline, all bacterial isolates were resistant to at least one antibiotic. Notably, Extended Spectrum Beta-Lactamase (ESBL) production was detected in 30% of Gram-negative isolates tested while 50% of the *Staphylococcus aureus* isolates tested positive for MRSA. There was a decline in the ESBL positive isolates over time (from 30% to 0) whereas MRSA positive isolates increased after treatment in the lesions (from 50% to 60%).

**Data availability statement:** All relevant data have been included in the manuscript and accompanying figures.

**Funding:** This work was funded on the BuruliNox study which is part of the EDCTP2 programme supported by the European Union (101897 BuruliNox TMA 2016 SF-1509 to ROP) and the Deutsche Forschungsgemeinschaft (DFG; Project number: 461611374 to TT, DD, YAA and ROP). The funders had no role in study design; in the collection, analysis and interpretation of data; in the writing of the report; and in the decision to submit the article for publication.

**Competing interests:** The authors have declared that no competing interests exist.

## Conclusions and significance

Results from this study highlight a concerning prevalence of antimicrobial resistant bacteria, including multi drug resistant (MDR), ESBL-positive and Methicillin-resistant *Staphylococcus aureus* (MRSA) pathogens, in Buruli ulcer lesions. These findings underscore the urgent need for the development of integrated guidelines to guide surveillance and treatment of secondary bacterial infections to further improve outcomes in BU.

## Author summary

Buruli ulcer (BU), a Neglected tropical Disease has been shown to harbour polymicrobial pathogens that can cause secondary infection and delay wound healing. Despite this, there is no guideline for managing these secondary infections and data on the antimicrobial resistance (AMR) of frequently isolated pathogens remain limited. To assess the AMR burden in BU, we longitudinally profiled antimicrobial resistance in patients from four endemic districts in Ghana using the VITEK 2. Our findings showed a concerning prevalence of multi drug resistant, ESBL and MRSA pathogens before, during and even after BU specific treatment. Notably, there was an increased presence of MRSA post BU treatment. These findings underscore the urgent need for the development of an integrated guideline on the surveillance and treatment of secondary infections in BU. Additionally, training programmes on wound management and infection control techniques for both healthcare providers and patients, and the provision of free dressing materials for patients could help mitigate the risk of wound infections. Further molecular studies are warranted to characterize resistance mechanisms and inform targeted interventions.

## Introduction

Antimicrobial resistance (AMR) has emerged as a major challenge to global public health threatening to undermine the effectiveness of current treatment regimens for a wide range of infectious diseases [1]. For Neglected Tropical Diseases (NTDs), the challenge may be particularly critical, given the increased antimicrobial use arising from preventive chemotherapy elimination and control programmes aimed at addressing these diseases [2–4]. In Sub-Saharan Africa where NTDs are prevalent, AMR could lead to increased disease burden, treatment failure and potentially hinder the progress made in NTD control particularly within these regions with limited healthcare access [2–5]. To ensure the continued efficacy of the currently used antimicrobials for NTDs and to realise the WHO NTD roadmap for 2030, it is essential to closely monitor drug efficacy, establish surveillance for monitoring resistance, define strategies to delay or curb drug resistance, and develop an appropriate arsenal of second-line treatments for NTDs. Nevertheless, there is limited information on AMR

in most NTDs. Among these diseases, AMR has been more widely studied in only six conditions: human African trypanosomiasis, leishmaniasis, onchocerciasis, schistosomiasis, soil-transmitted helminths, and trachoma [2].

Buruli ulcer (BU) is a necrotizing skin condition caused by *Mycobacterium ulcerans*, which can lead to functional disfigurement and disability as well as psychosocial consequences in the absence of prompt, appropriate medical treatment [6–8]. Classified as an NTD, the disease frequently occurs in impoverished rural areas of West Africa, but there has been a recent surge in cases from Australia [9]. BU presents as painless nodules, plaques and oedematous forms, which can progress to extensive ulcers and occasionally osteomyelitis. Disease transmission remains an enigma despite major advances in understanding the mechanism of BU. Regarding treatment, good outcomes with significant reductions in recurrence rates from as high as 28% to 1.44% have been observed since the shift from surgery to antibiotics [10–12]. Currently, the recommended treatment involves a full oral combination of rifampicin and clarithromycin for 56 days [13,14]. Although the current regimen is effective, some patients suffer substantial delays in healing, which have mainly been attributed to *M. ulcerans* and mycolactone persistence [15], paradoxical reactions [16] and secondary infections with pathogenic organisms [17,18].

There is a practice of prescribing antibiotics for suspected secondary bacterial infection in BU [18]; however, there are no guidelines for managing these infections [18], and this could contribute to the growing menace of AMR. Opportunistic pathogens such as *Staphylococcus aureus, Pseudomonas aeruginosa, Klebsiella pneumoniae*, and *Escherichia coli* have often been isolated from these lesions [17,19–21]. There are reports that bacterial isolates from BU lesions from West Africa have resistance to commonly prescribed first-line antibiotics for infectious diseases [19,21–23]. Of particular concern is the growing presence of multidrug-resistant (MDR) organisms, including *Extended Spectrum Beta-Lactamase* (ESBL) producing bacteria and *Methicillin-resistant Staphylococcus aureus* (MRSA) in BU lesions [17,19,21,23–25]. Bacteria that produce ESBLs and MRSA strains continue to pose significant public health risks as they resist many commonly prescribed antibiotics, making infections difficult to treat. Additionally, the presence of multidrug-resistant organisms, even after treatment, may facilitate the transfer of resistance genes among species complicating treatment efforts [26]. In 2019, *Staphylococcus aureus, Klebsiella pneumoniae,* and *Escherichia coli*, were linked to AMR-related fatalities on a global scale [27].

Given the frequent isolation of these pathogens in BU lesions and their association with AMR fatalities, it is vital to understand their impact on BU management, particularly since there are currently no treatment guidelines for managing secondary bacterial infections. We aimed to describe the patterns of antimicrobial resistance in pathogenic bacterial isolates from BU lesions.

## Methods

### Ethics statement

This study was approved by the Committee on Human Research, Publication and Ethics (CHRPE) of the Kwame Nkrumah University of Science and Technology (KNUST) with approval numbers CHRPE/AP/472/17 and CHRPE/AP/608/22. All participants provided written informed consent. For young children (<18 years), written consent was obtained from parents or legal guardians. Individuals aged 11 – 17 years provided assent in addition to informed consent obtained from their parents or legal guardians. All study procedures conformed with the principles guiding research in human subjects as set out in the Declaration of Helsinki [28].

### Study design, study area and recruitment

A longitudinal study approach was utilized to assess AMR in bacterial isolates from the lesions of patients with BU between August 2021 and June 2024. Participants were recruited from four BU endemic districts in Ghana (Fig 1): Agogo Presbyterian Hospital in the Asante Akim North District, Tepa Government Hospital in the Ahafo Ano North District,



**Fig 1. Map of Ghana showing the study sites.** Map was generated using Quantum GIS version (QGIS) QGIS 3.34 LTR (Prizren). The shapefiles for Ghana and the various regions obtained from OpenStreetMap (https://www.openstreetmap.org/copyright, CC BY-SA 2.0) were utilized as data sources for plotting the map. Map data from OpenStreetMap https://www.openstreetmap.org/copyright.

Dunkwa Government Hospital in the Upper Denkyira East District and Wassa Akropong Municipal Hospital in Wassa Amenfi East Municipality. These hospitals have established treatment centres for BU management within the district and neighbouring communities. Participants were selected using a convenience sampling technique. The aims and study procedures were explained to participants and consent was obtained prior to recruitment. Participants were included if they were; PCR positive for BU, aged between 5 and 80 years, willing to provide consent, in generally good health, and not requiring any long-term medication. Pregnant or breastfeeding women, patients who were already undergoing antibiotic therapy for BU, and those with chronic ulcers unrelated to BU were excluded.



### Laboratory procedures

**Confirmation of suspected BU.** Laboratory confirmation of clinically suspected BU cases by IS*2404* qPCR was conducted according to standard protocol in the laboratory of the Kumasi Centre for Collaborative Research (KCCR) as detailed elsewhere [15,16]. Following confirmation of BU, wound swabs for bacterial culture were collected at baseline (prior to the initiation of antibiotic treatment), and at weeks 8 and 16 if the lesion had not healed. At each sampling time, sterile swab sticks moistened with phosphate-buffered saline were used to swab the surface of the ulcer lesions following the Levine sample collection method [29]. Briefly, samples were collected by rotating the tip of the sterile swab over a $1\,cm^2$ area of wound surface while exerting sufficient pressure to extract fluid from the wound tissue. For nodules, oedematous and plaque lesions that had began to ulcerate, samples were collected from the ulcerated surfaces of these lesions. Samples were kept on ice at $4–8^0C$ and transported to the KCCR laboratory within an average time frame of 3 hours for further processing. The samples were processed immediately upon arrival at the laboratory.

**Bacterial culture, identification and Antibiotic susceptibility testing (AST).** Upon arrival at the laboratory, samples were pre-enriched in Brain Heart Infusion broth (BD, USA) for 24 hours and then cultured on MacConkey agar, Blood agar at 37°C for 18–24 hours. The initial identification of the bacteria was based on their morphology, colour and reaction to the agar used. Sub-cultures were prepared consecutively until pure isolates of *Staphylococcus aureus*, *Pseudomonas aeruginosa*, *Escherichia coli*, *Klebsiella pneumoniae* were obtained. The colonies of these isolates were further identified using their Gram status and their reactions to biochemical tests including catalase, coagulase, oxidase and indole tests. Subsequently, the colonies were confirmed using the VITEK 2 compact system (Biomerieux, France) according to the manufacturer's instructions. Colonies of *Providencia stuartii* and *Enterococcus* species were included in this study as they have also been frequently isolated from BU lesions [18].

Antimicrobial susceptibility testing of bacterial isolates was performed following the manufacturer's instruction (Bio-merieux, Marcy-l' Étoile, France). Gram negative bacteria were tested using the AST N214 cards while the Gram-positive cocci were tested using the AST GP 67 cards. The minimum inhibitory concentration (MIC) results were interpreted using the Advanced Expert System employing the European Committee on Antimicrobial Susceptibility Testing (EUCAST) guidelines version 2017 (www.eucast.org).

### Data analysis

Data generated from the study was entered into Microsoft Excel version 2021 (Microsoft Corporation, Redmond, WA, USA) and analysed using GraphPad Prism version 9.0 (GraphPad Software, Inc., La Jolla, CA, USA). Study data was described using frequencies and percentages.

## Results

### Study population

A total of 56 patients were recruited and enrolled in the study. Among these, 33 (59%) were males. The median age of participants was 22.5 years (IQR: 10.8 – 45.3). The most common clinical lesion form was the ulcer (46/56, 82.1%). Twenty-four of the 56 participants (43.6%) presented with category III lesions. Most lesions were located on the lower limbs (38/56, 67.9%). The demographic and clinical characteristics of the participants are detailed in Table 1.

### Distribution of pathogenic bacteria with treatment

The details of the workflow for samples collected from PCR confirmed BU participants for bacterial culture is shown in Fig 2. A total of 169 bacteria isolates from eight species (including 6 Gram negative and 2 Gram positive organisms) were obtained from the lesions of the participants. There were 76 isolates at baseline, 43 at week 8 and 50 at week 16.

The distribution of commonly isolated bacteria in BU lesions identified in this study cohort is illustrated in Fig 3. At baseline, *E. faecalis* (19/76, 25%) was the most dominant organism. The common organisms at week 8 included

**Table 1. Demographic and clinical characteristics of participants at baseline.**

| Characteristic | Frequency, n (%) |
|---|---|
| **Sex** | |
| Male | 33 (58.9) |
| Female | 23 (41.1) |
| **Age (years)** | 22.5 (10.8 – 45.3) * |
| ≤15 | 22 (39.3) |
| 16-29 | 12 (21.4) |
| 30-49 | 13 (23.2) |
| ≥50 | 9 (16.1) |
| **Occupation** | |
| Farmer | 17 (30.3) |
| Student | 26 (46.4) |
| Artisans, teachers and traders | 8 (14.3) |
| Miner | 3 (5.4) |
| Unemployed | 2 (3.6) |
| **Clinical form** | |
| Ulcer | 46 (82.2) |
| Plaque | 4 (7.1) |
| Nodule | 4 (7.1) |
| Oedema | 2 (3.6) |
| **Lesion location** | |
| Upper Limb | 14 (25) |
| Lower Limb | 38 (67.9) |
| Others⁺ | 4 (7.1) |
| **Lesion category** | |
| Category I (< 5 cm in diameter) | 15 (26.7) |
| Category II (5 – 15 cm in diameter) | 17 (30.4) |
| Category III (˃ 15 cm in diameter) | 24 (42.9) |

*Median age (IQR), + Head and neck region, back, groin

Nodules, oedematous and plaque lesions had began to ulcerate

*E. faecalis* (12/43, 28%), *P. mirabilis* (7/43, 16.3%) and *P. stuartii* (7/43, 16.3%). The predominant isolates from lesions at week 16 were *S. aureus* (13/50, 26%), *E. faecalis* (10/50, 20%) and *P. mirabilis* (10/50, 20%). Overall, there was a trend of decrease in bacteria frequency during treatment (weeks 0–8). After week 8, the proportion of *S. aureus* (from 3/43–13/50) and *P. mirabilis* (from 7/43–10/50) increased.

### Pattern of antibiotic resistance profile among bacterial isolates

Utilizing the VITEK 2 system, susceptibility tests were conducted on a total of 166 of the 169 bacterial isolates comprising 93 Gram-negative bacilli and 73 Gram-positive cocci. AST results were not generated for the three isolates as a result of technical issues.

### Resistance profile of Gram-negative bacteria

Table 2 presents the resistance profile of Gram-negative bacteria over time. Bacterial isolates exhibited varying patterns of resistance to the selected antibiotics both before and after treatment. High levels of resistance to ampicillin/sulbactam



**Fig 2. Workflow for antimicrobial resistant bacteria isolated from Buruli ulcer lesions.** Details of the workflow for samples collected from PCR confirmed BU participants for bacterial culture. Swab samples were collected from each lesion (patient) at time points 0 (baseline; before initiation of antibiotic treatment), Week 8 (8 weeks after treatment initiation), and week 16 (16 weeks after treatment initiation). Week 8: 48 individuals were sampled for bacterial culture (8 participants had lesions that were healed). Week 16: 43 individuals were sampled for bacterial culture (5 more participants had lesions that were healed).

were noted in *Enterobacter* spp. (10/10, 100%), *P. stuartii* (6/6, 100%) and *Klebsiella* spp. (2/4, 50%) at baseline. Similarly, organisms showed high resistance to tetracycline and trimethoprim/sulfamethoxazole at baseline. Overall, there was a trend of decreasing resistance to the antibiotics by week 8. However, *Enterobacter* species remained consistently resistant to ampicillin/ sulbactam (6/6, 100%) and *P. stuartii* exhibited was high resistance to tetracycline (6/6, 100%). At week 16, a higher proportion of *Klebsiella* spp. (1/2, 50%), *P. mirabilis* (9/10, 90%) and *P. stuartii* (3/4, 75%) showed resistance to trimethoprim/sulfamethoxazole compared to week 8. All isolates continued to be susceptible to meropenem throughout the study period.

**Resistance profile of Gram-positive bacteria**

Among Gram-positive bacteria, there was no resistance of *E. faecalis* isolates to all tested antibiotics except to tigecycline (1/19, 5.3%) and nitrofurantoin (2/19, 10.5%) at baseline. By the end of week 8, no isolates (0/14) demonstrated resistance to the tested antibiotics. At week 16, *E. faecalis* exhibited resistance only to nitrofurantoin (1/10, 10%). Throughout

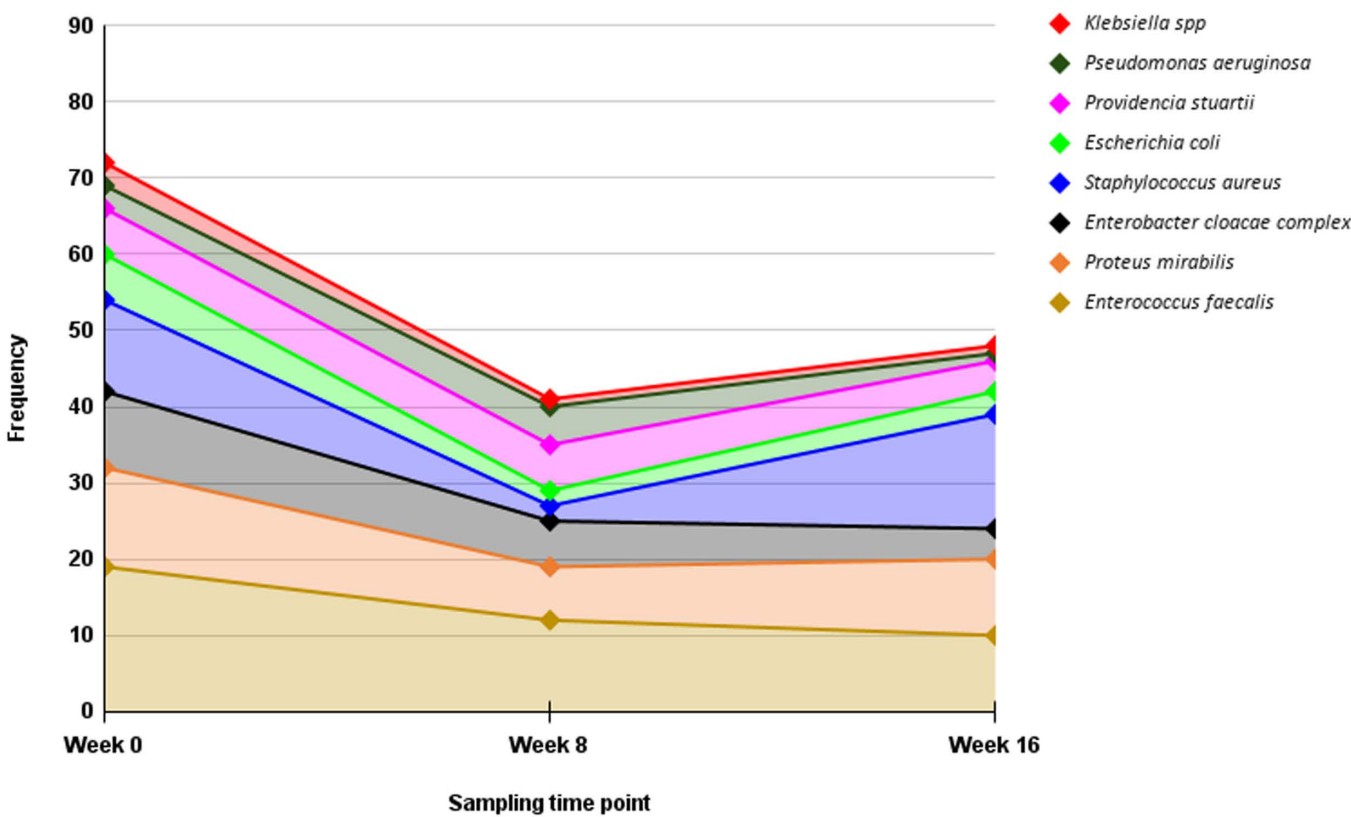

**Fig 3. Distribution of commonly reported bacterial isolates in Buruli ulcer lesions.** The y-axis represents the frequency (number) of isolates, while the x-axis indicates the sampling time. Each coloured line and shaded region represent specific bacterial species showing changes in bacterial prevalence over time. Klebsiella spp. refers to Klebsiella pneumoniae, and Klebsiella oxytoca.

the study, *E. faecalis* species isolated from this cohort did not show any resistance to ampicillin, linezolid and vancomycin at all time points.

Among the *S. aureus* isolates, high resistance to benzylpenicillin (92%) and tetracycline (83.3%) was demonstrated at baseline. All isolates were susceptible to clindamycin, quinupristin/dalfopristin, linezolid, vancomycin and tigecycline both before and after the BU recommended treatment (week 8). Furthermore, the susceptibility of *S. aureus* isolates to linezolid, quinupristin/dalfopristin and tigecycline was noted at week 16. However, a higher proportion of the bacterial isolates at week 16 exhibited resistance to the majority of the selected antibiotics, including oxacillin (9/15, 60%), tetracycline (14/15, 93.3%) and rifampicin (2/2, 100%) compared to before treatment. The proportion of *S aureus* isolates that showed resistance to macrolides like erythromycin during treatment was relatively low. One of two (50%) *S aureus* isolates were resistant to rifampicin at week 0 while 2/2 (100%) were resistant at week 16. Table 3 presents the resistance profile of Gram-positive bacteria to selected antibiotics at various study time points.

### Distribution of phenotypic ESBL and MRSA isolates in BU lesions using VITEK

Table 4 shows the frequency of ESBL and MRSA isolates across the study period. The distribution of ESBL-positive bacteria reduced from before treatment (3/10, 30%) to post-treatment (0/5, 0%). Among the *S. aureus* isolates screened, the distribution of MRSA increased post-treatment (9/15, 60%). MRSA isolates were susceptible to quinupristin-dalfopristin, linezolid and tigecycline. Only one MRSA was vancomycin resistant.

**Table 2. Resistance profile of Gram-negative bacteria.**

| Antibiotic | Week of test | *Enterobacter spp*, n (%) | *Escherichia coli*, n (%) | *Klebsiella spp*, n (%) | *Proteus mira-bilis*, n (%) | *Providencia stuartii*, n (%) | *Pseudomonas aeruginosa*, n (%) |
|---|---|---|---|---|---|---|---|
| Ampicillin/sulbactam | 0 | 10/10 (100) | 2/6 (33.3) | ND | 1/13(7.7) | 6/6 (100) | ND |
| | 8 | 6/6 (100) | 1/2(50) | ND | 0/7 (0) | 5/6 (83.3) | ND |
| | 16 | 4/4 (100) | 0/3 | ND | 4/10 (40) | 4/4(100) | ND |
| Piperacillin/Tazobactam | 0 | 10/10 (100) | 1/6 (16.7) | 2/4 (50) | 0/13 | 0/6 | 1/3 (33.3) |
| | 8 | 0/6 | 1/2 (50) | 0/1 | 0/7 | 0/6 | 0/5 |
| | 16 | 0/4 | 0/3 | 0/2 | 0/10 | 0/4 | 0/1 |
| Cefuroxime | 0 | 7/10 (70) | 1/6 (16. 7) | 2/4(50) | 1/13 (7.7) | 6/6 (100) | ND |
| | 8 | 3/6 (50) | 1/2(50) | 0/1 | 0/7 | 5/6 (83.3) | ND |
| | 16 | 3/4 (75) | 0/3 | 0/2 | 0/10 | 4/4(100) | ND |
| Cefpodoxime | 0 | 7/10 (70) | 1/6 (16.7) | 2/4(50) | 0/13 | 0/6 | ND |
| | 8 | 2/6 (33.3) | 1/2 (50) | 0/1 | 0/7 | 0/6 | ND |
| | 16 | 3/4 (75) | 0/3 | 0/2 | 0/10 | 0/4 | ND |
| Meropenem | 0 | 0/8 | 0/6 | 0/4 | 0/13 | 0/6 | 0/3 |
| | 8 | 0/6 | 0/2 | 0/1 | 0/7 | 0/6 | 0/5 |
| | 16 | 0/4 | 0/3 | 0/2 | 0/10 | 0/4 | 0/1 |
| Gentamicin | 0 | 0/10 | 0/6 | 1/4 (25) | 2/13 (15.4) | 6/6 (100) | 0/3 |
| | 8 | 0/6 | 0/2 | 0/1 | 0/7 | 5/6 (83.3) | 0/5 |
| | 16 | 0/4 | 0/3 | 0/2 | 2/10 (20) | 4/4 (100) | 0/1 |
| Moxifloxacin | 0 | 4/10 (40) | 0/6 | 1/4 (25) | 2/13 (15.4) | 0/6 | ND |
| | 8 | 0/6 | 0/2 | 0/10 | 0/7 | 3/6 (50) | ND |
| | 16 | 0/4 | 0/3 | 0/2 | 0/10 | 1/4 (25) | ND |
| Tetracycline | 0 | 5/10 (50) | 4/6 (66.7) | 2/4 (50) | 12/13 (92.3) | 6/6 (100) | ND |
| | 8 | 1/6 (16.7) | 2/2(100) | 0/1 | 1/7 (14.3) | 6/6 (100) | ND |
| | 16 | 0/4 | 1/3(33.3) | 1/2(50) | 10/10 (100) | 4/4 (100) | ND |
| Trimethoprim/ Sulfamethoxazole | 0 | 5/10 (50) | 3/6 (50) | 2/4(50) | 7/13(53.9) | 4/6 (66.7) | ND |
| | 8 | 1/6(16.7) | 2/2 (100) | 0/1(0) | 6/7 (85.7) | 2/6 (33.3) | ND |
| | 16 | 1/4 (25) | 0/3 | 1/2(50) | 9/10 (90) | 3/4 (75) | ND |

*Numerator represents number of isolates that were resistant to a given antibiotic; denominator represents the total number of isolates tested at the time point. ND means not done.

## Discussion

Chronic ulcers including BU present significant challenges in the global fight against AMR due to their polymicrobial nature, prolonged treatment courses, and role as reservoirs for resistant pathogens. We found the presence of pathogenic bacteria with varying levels of resistance to commonly used antibiotics in BU lesions before, during and after BU-specific antibiotic treatment. These findings provide critical insights into the local resistance patterns associated with BU in Ghana and data to guide therapeutic strategies.

In this study cohort, the median age of BU patients was slightly higher than previously reported [9,15,16]. The slightly higher median age observed here may be attributed to the changing epidemiology of the disease worldwide. However, most participants presented with larger lesions (category II and III), predominantly ulcers on the lower limbs in line with the known epidemiology of the disease in West Africa [30,31].

We previously established that higher bacterial diversity of BU colonizers influences healing time in diseased persons and MDR organisms were predominantly found in slow healing lesions [17]. In this study, we focused on the resistance

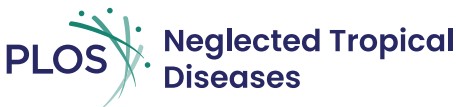

**Table 3. Resistance profile of Gram-positive bacteria across study weeks.**

| Organism | Antibiotics | Week 0, (%) | Week 8, (%) | Week 16, (%) |
|---|---|---|---|---|
| *Enterococcus faecalis* | Ampicillin | 0/19 | 0/14 | 0/11 |
| | Linezolid | 0/19 | 0/14 | 0/11 |
| | Vancomycin | 0/19 | 0/14 | 0/11 |
| | Tigecycline | 1/19 (5.3) | 0/14 | 0/11 |
| | Nitrofurantoin | 2/19 (10.5) | 0/14 | 1/10 (10) |
| *Staphylococcus aureus* | Benzylpenicillin | 11/12 (92) | 2/2 (100) | 15/15 (100) |
| | Oxacillin | 6/12 (50) | 1/2 (50) | 9/15 (60) |
| | Gentamicin | 1/12 (8.3) | 0/2 | 2/15 (13.3) |
| | Ciprofloxacin | 1/12 (8.3) | 0/2 | 2/15 (13.3) |
| | Levofloxacin | 1/12 (8.3) | 0/2 | 2/15 (13.3) |
| | Moxifloxacin | 1/12 (8.3) | 0/2 | 1/15 (6.7) |
| | Erythromycin | 1/12 (8.3) | 0/2 | 2/15 (13.3) |
| | Clindamycin | 0/12 | 0/2 | 3/15 (20) |
| | Quinupristin/Dalfopristin | 0/12 | 0/2 | 0/15 |
| | Linezolid | 0/12 | 0/2 | 0/15 |
| | Vancomycin | 0/12 | 0/2 | 1/15 (6.7) |
| | Tetracycline | 10/12 (83.3) | 1/2 (50) | 14/15 (93.3) |
| | Tigecycline | 0/12 | 0/2 | 0/15 |
| | Nitrofurantoin | 0/2 | 0 | ND |
| | Rifampicin | 1/2 (50) | 0 | 2/2 (100) |
| | Trimethoprim/Sulfamethoxazole* | 1/11 (9.1) | 0/2 | 5/15 (33.3) |

Numerator represents number of isolates that were resistant to a given antibiotic; denominator represents the total number of isolates tested at the time point. ND means not done.

**Table 4. Distribution of ESBL and MRSA isolates in lesions of Buruli ulcer after the initiation of treatment.**

| | ESBL positive, (%) | MRSA, (%) |
|---|---|---|
| Week 0 | 3/10 (30) | 6/12 (50) |
| Week 8 | 1/3 (33.3) | 1/2 (50) |
| Week 16 | 0/5 (0) | 9/15 (60) |

patterns of pathogenic isolates over the treatment period. At baseline, *Enterococcus faecalis* and *Proteus mirabilis* were the dominating isolates in this study. The study by Yeboah-Manu et al [21] identified *S. aureus*, *P. aeruginosa* and *P. mirabilis* as the dominating bacteria at baseline while Barogui et al [19] reported *S. aureus* and *P. aeruginosa* as the dominant isolates. The differences in dominant isolates at baseline across these studies may be attributed to geographical differences, the sampling techniques used, or the various methods of bacterial identification employed.

There was a general decline in the prevalence of all pathogens, except *P. aeruginosa* and *P. stuartii* during the 8-week treatment period consistent with reports from a previous longitudinal study on the bacterial burden in BU lesions [20]. The decline observed here may be attributed to the administration of the broad-spectrum antibiotic treatment over the 56 days period. Remarkably, there was an increase in the prevalence of multi drug resistant (MDR) pathogens, particularly *S. aureus* isolates after the treatment duration. Several factors may have contributed to this worrying trend. Firstly, the continuous administration of clarithromycin and rifampicin antibiotics over the long treatment period may have led to the clearance of susceptible isolates allowing MDR strains to persist and proliferate. This observation may also be due to

induction of resistance in these organisms. Resistance may have been induced in bacterial strains after exposure to the BU specific antibiotics through DNA mutations or horizontal gene transfer [32,33]. Another possible explanation for the surge in resistant pathogens could be the contamination of wounds during dressing changes. The median healing time of most BU lesions has been documented between 8 and 48 weeks [34–38]. During this period, there is a possibility of nosocomial acquisition of MDR pathogens in the wounds of these patients through inadequate hygiene as documented in previous studies [23,39,40]. Preventive measures including proper training in wound management and infection control techniques, along with the provision of adequate dressing materials, can help reduce the risk of nosocomial resistant pathogens in BU wounds.

In addition, BU specific antibiotics (Clarithromycin, rifampicin) are provided freely for BU patients under the auspices of the Buruli ulcer National Control programme. However, patients must cover the cost of dressings and other materials during wound management. Typically, BU affects the poor within these rural endemic regions who may not be able to procure new dressing materials, thus tend to wash and recycle their dressings. High bacterial load has been isolated and enumerated from these recycled bandages highlighting their role in wound contamination [21]. Presently, the recommended dressing for BU management involves the use of vaseline gauze dressings which have no added advantage on secondary organisms within BU wounds. Provision of free bandages to BU patients and the adoption of advanced dressing materials such as nitric oxide dressings and dialkylcarbamoyl chloride (DACC) coated dressings could go a long way in reducing the risk of secondary infection and subsequently combating antibiotic resistance [41,42].

Bacteria isolates were subjected to susceptibility testing to antimicrobials before and after treatment completion. Before treatment, most Gram-negative isolates showed varying levels of resistance to all antibiotics tested except for the carbapenem group. The highest levels of resistance were observed against tetracyclines. High levels of resistance to ampicillin/sulbactam, tigecycline and trimethoprim-sulfamethoxazole were also observed. This observation is worrying considering these drugs are commonly used in the treatment of infections in Ghana [18,43]. After the treatment (week 16), Gram-negative bacteria demonstrated little to no resistance to the majority of the antibiotics initially tested at baseline. Resistance to tetracycline on the other hand was consistently high, even peaking at 100% for some isolates at week 16. The observation of high resistance to tetracycline is not surprising as it is widely used in the treatment of infections in both humans and animals. In 2011, Newman et al reported a high resistance of 82% to tetracycline in different clinical samples across various hospitals in Ghana [43]. Among Gram positive bacteria, *E. faecalis* and *S. aureus* showed varying patterns of resistance to the selected antibiotics. *Staphylococcus aureus* isolates particularly showed increased resistance to the beta-lactam, aminoglycosides, fluoroquinolones, tetracycline, rifamycin and sulfonamide class of antibiotics after BU treatment. This trend after the BU treatment duration probably stems from its ability to harbour virulence factors and other genetic mobile elements known to promote immune evasion and development of superantigens necessary for its survival [44]. All *Staphylococcus* isolates remained susceptible to the tigecycline and linezolid. Although tigecycline and linezolid are already registered, their limited availability hinders their use.

The phenotypic ESBL producing strains decreased significantly throughout the study period, in keeping with the reduced burden of resistant Gram-negative bacteria within the wounds. Unlike ESBLs, the distribution of MRSA phenotypic strains was high after treatment (60%). Similar to previous observations [19,21], the distribution of *S. aureus* isolates was low during the treatment period suggesting the increased distribution in phenotypic MRSA strains after the BU treatment duration may have been hospital or community acquired during wound dressings. MRSA isolates were however susceptible to quinupristin-dalfopristin, linezolid and tigecyclyine. As indicated earlier, the cost of BU treatment in Ghana is free; however other ancillary costs including antibiotics for secondary infection among others remain the responsibility of the patient and their caregivers [6,45]. Thus, the rise in MRSA following BU treatment necessitates urgent and robust infection control measures in both clinical and community settings.

To join the global fight against AMR especially in resource limited settings for BU, the importance of expanding antimicrobial stewardship programmes as a tool for combating secondary bacterial infections in BU cannot be overstated.

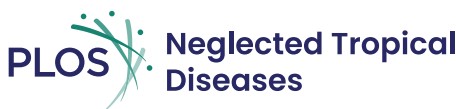

Routine antimicrobial susceptibility testing and the inclusion of resistance surveillance in the national BU management protocols, aligned with the national AMR action plan could significantly enhance clinical outcomes and support global efforts to tackle the menace of AMR. While providing free antibiotics remains the cornerstone of Buruli ulcer treatment, ensuring access to free wound dressing materials for BU patients could further improve clinical outcomes by facilitating optimal wound care and minimising secondary infections.

Overall, this study has provided further evidence of the broader issue of AMR in resource limited countries and emphasizes the necessity for effective antimicrobial stewardship to mitigate the scourge of AMR. Nevertheless, there were several limitations. Firstly, while the sample size was appreciable, it was restricted to locations within the middle belt of Ghana. Although we observed an increase in the proportion of pathogenic organisms such as MRSA and *Proteus mirabilis* that were resistant to the tested antibiotics, the overall numbers were relatively low. Thus, similar studies in other endemic settings involving larger number of participants are needed to corroborate these findings. Furthermore, Whole Genome Sequencing which could have provided deeper insights into the genetic drivers of resistance and indicated if in-hospital transmission occurred was not performed. We also did not quantify bacterial load, correlate it with clinical signs of infection, or conduct histopathological analysis to confirm secondary infection. Lastly, as treatment was not guided by AST results, we could not assess its impact on clinical outcomes. Therefore, we recommend that future studies address these issues to provide more insight into the burden of AMR in BU.

## Conclusion

There is a concerning prevalence of antimicrobial resistant bacteria, including MDR, ESBL-positive and MRSA pathogens, in Buruli ulcer lesions. These findings underscore the urgent need for the development of integrated guidelines to guide surveillance and treatment of secondary bacterial infections to further improve outcomes in BU. To maximise usefulness, local settings will need to conduct surveillance to provide appropriate microbiological data to support the development of local antibiotic guidelines.

Additionally, the development of tailored and context specific therapeutic strategies to address bacteria resistance challenges and optimize patient care in BU cannot be overlooked.

## Acknowledgments

We are grateful to all study participants and network of staff at the BU clinics in the selected study sites. We also thank the Skin NTD research group, the Bacteriology team at KCCR, Mr. Richmond Gorman, Mr. Louis Kyei-Tuffour and Dr Jonathan Kofi Adjei for their diverse contribution to this study.

## Author contributions

**Conceptualization:** Nancy Ackam, Augustina Sylverken, Kwasi Obiri-Danso, Mark Wansbrough-Jones, Thorsten Thye, Denise Dekker, Yaw Ampem Amoako, Richard Odame Phillips.

**Data curation:** Nancy Ackam, Abigail Opoku Boadi, Abigail Agbanyo, Yaw Ampem Amoako.

**Formal analysis:** Nancy Ackam, Yaw Ampem Amoako.

**Funding acquisition:** Mark Wansbrough-Jones, Thorsten Thye, Denise Dekker, Yaw Ampem Amoako, Richard Odame Phillips.

**Investigation:** Nancy Ackam, Abigail Opoku Boadi, Abigail Agbanyo.

**Methodology:** Nancy Ackam, Charity Wiafe Akenten, Kwasi Obiri-Danso, Mark Wansbrough-Jones, Thorsten Thye, Denise Dekker, Yaw Ampem Amoako, Richard Odame Phillips.

**Project administration:** Nancy Ackam, Abigail Agbanyo, Yaw Ampem Amoako.

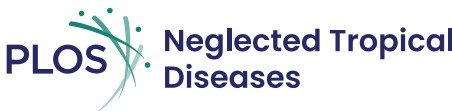

**Resources:** Kabiru Mohammed Abass, George Amofa, Elizabeth Ofori, Joseph Azabire, Mark Wansbrough-Jones, Thorsten Thye, Denise Dekker, Richard Odame Phillips.

**Supervision:** Charity Wiafe Akenten, Augustina Sylverken, Kwasi Obiri-Danso, Mark Wansbrough-Jones, Yaw Ampem Amoako, Richard Odame Phillips.

**Validation:** Charity Wiafe Akenten, Kabiru Mohammed Abass, George Amofa, Elizabeth Ofori, Joseph Azabire, Augustina Sylverken, Kwasi Obiri-Danso, Mark Wansbrough-Jones, Thorsten Thye, Denise Dekker, Yaw Ampem Amoako, Richard Odame Phillips.

**Visualization:** Charity Wiafe Akenten, Abigail Opoku Boadi, Abigail Agbanyo, Kabiru Mohammed Abass, George Amofa, Elizabeth Ofori, Joseph Azabire.

**Writing – original draft:** Nancy Ackam, Yaw Ampem Amoako.

**Writing – review & editing:** Nancy Ackam, Charity Wiafe Akenten, Abigail Opoku Boadi, Abigail Agbanyo, Kabiru Mohammed Abass, George Amofa, Elizabeth Ofori, Joseph Azabire, Augustina Sylverken, Kwasi Obiri-Danso, Mark Wansbrough-Jones, Thorsten Thye, Denise Dekker, Yaw Ampem Amoako, Richard Odame Phillips.

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
