## [Decision Letter · Decision Letter 0]

6 Apr 2025

PNTD-D-25-00354

Antimicrobial resistant bacteria isolated from Buruli ulcer lesions in Ghana

Dear Dr. Amoako,

Thank you for submitting your manuscript to PLOS Neglected Tropical Diseases. After careful consideration, we feel that it has merit but does not fully meet PLOS Neglected Tropical Diseases's publication criteria as it currently stands. Therefore, we invite you to submit a revised version of the manuscript that addresses the points raised during the review process.

Please submit your revised manuscript within 60 days Jun 05 2025 11:59PM. If you will need more time than this to complete your revisions, please reply to this message or contact the journal office at plosntds@plos.org. Please include the following items when submitting your revised manuscript:

We look forward to receiving your revised manuscript.

Kind regards,

Paul J. Converse

Academic Editor

Stuart Blacksell

Section Editor

Shaden Kamhawi

co-Editor-in-Chief

Paul Brindley

co-Editor-in-Chief

**Additional Editor Comments:**

Dear Dr. Amoako,

Your manuscript has been reviewed by three experts in the field. The reviewers found that, with revisions, the paper would add to the knowledge of bacterial co-colonization in Buruli ulcer lesions. Overall, please revise to make the manuscript more concise as well as to address all of the reviewers' concerns. If you choose to resubmit your manuscript, please address all of the concerns in response to each point in the reviews, particularly those pointed out by Reviewer 1 and the issue identified by Reviewers 2 and 3, with both a changes marked version and a clean version. Thank you for submitting your manuscript to PLoS Neglected Tropical Diseases.

**Journal Requirements:**

1) Some material included in your submission may be copyrighted. According to PLOSu2019s copyright policy, authors who use figures or other material (e.g., graphics, clipart, maps) from another author or copyright holder must demonstrate or obtain permission to publish this material under the Creative Commons Attribution 4.0 International (CC BY 4.0) License used by PLOS journals. Please closely review the details of PLOSu2019s copyright requirements here: PLOS Licenses and Copyright. If you need to request permissions from a copyright holder, you may use PLOS's Copyright Content Permission form.

Potential Copyright Issues:

i) Figure 1. Please (a) provide a direct link to the base layer of the map (i.e., the country or region border shape) and ensure this is also included in the figure legend; and (b) provide a link to the terms of use / license information for the base layer image or shapefile. We cannot publish proprietary or copyrighted maps (e.g. Google Maps, Mapquest) and the terms of use for your map base layer must be compatible with our CC BY 4.0 license.

2) We note that your Data Availability Statement is currently as follows: "All relevant data have been included in the manuscript.". Please confirm at this time whether or not your submission contains all raw data required to replicate the results of your study. Authors must share the “minimal data set” for their submission. PLOS defines the minimal data set to consist of the data required to replicate all study findings reported in the article, as well as related metadata and methods (https://journals.plos.org/plosone/s/data-availability#loc-minimal-data-set-definition).

3) Please amend your detailed Financial Disclosure statement. This is published with the article. It must therefore be completed in full sentences and contain the exact wording you wish to be published.

1) State the initials, alongside each funding source, of each author to receive each grant. For example: "This work was supported by the National Institutes of Health (####### to AM; ###### to CJ) and the National Science Foundation (###### to AM).".

**Reviewers' Comments:**

Reviewer's Responses to Questions

**Key Review Criteria Required for Acceptance?**

**Methods**

-Are the objectives of the study clearly articulated with a clear testable hypothesis stated?

-Is the study design appropriate to address the stated objectives?

-Is the population clearly described and appropriate for the hypothesis being tested?

-Is the sample size sufficient to ensure adequate power to address the hypothesis being tested?

-Were correct statistical analysis used to support conclusions?

-Are there concerns about ethical or regulatory requirements being met?

Reviewer #1: Methods do not entirely cover what is needed to fully address the objectives of the study. Although the Introduction seems to promiss a lot of novel information on the subject - the role of 'secondary infection' or perhaps, rather - the role of bacterial colonisation of BU wounds - on healing (i.e., impaired or delayed healing) of such wounds; - the last sentence of the Introduction Section seems to be much more modest i.e., describing colonisaton with potentially pathogeneic bacteria that MIGHT delay or impair BU wound healing. The authors provide a nice review of the literature on this subject; it remains unclear however, what information they plan to add to the existing literature. The Introduction promises a confirmation that bacterial secondary infection drives / causes delayed wound healing; but obviously this hypothesis cannot be proved by providing associations between delayed healing and bacterial colonisation; cause and effect cannot be proven using such methodology, without a prospective, well powered randomized intervention, associations do not prove cause and effect (ref 17 shows association, not causation). An intersting aspect of the study was that culture samples were obtained at three different points over time. One weakness is obviously, the fact that only isolates were obtained by culture; if genomic methods had been used with quantification, perhpas a much more realistic reflection of potential bacterial pathogens would have been generated.

Even if that be the case, the methods are not entirely clear;

1) although selection criteria were described for study participants, no sample size estimation was provided; was it rather a convenience sample? Why stop enrolling after 56 particpants were enrolled? Could the authors provide a flow chart for readers to understand how many screened patients were excluded, and for what reasons?

2) how exactly did they swab the lesions? Amissah et al used a strictly defined method to sample the whole surface of lesions using the bandages of lesions removed during wound care and dressing changes; other study groups sampled by swab using aless clearly defined method, obviously intruducing noise and perhaps even, bias in the study methodology; please explain the sampling in more detail.

3) although most lesions were ulcers or plaque, it puzzeled this reviewer how the authors sampled nodules and edematous lesions; or did these lesions break down into ulcers later on, and were the samples only taken after the lesions broke down into ulcers?

Reviewer #2: Yes

Reviewer #3: Study design was appropriate but there are errors/omissions in the statistical analysis as outlined below.

**Results**

-Does the analysis presented match the analysis plan?

-Are the results clearly and completely presented?

-Are the figures (Tables, Images) of sufficient quality for clarity?

Reviewer #1: The Results summarize the findings obtained from the 56 study participants, with a total of 196 isolates of bacterial species and their antimicrobial resistance using EUCAST criteria for breakpoints, with most isolates obtained at baseline; 76 isolates at baseline, 43 at week 8 and 50 at week 16. Although table 1 summarizes the data on the study subjects, many different aspects that have potential impact on wound healing are not provided; nutritional status, duration of disease, exact lesional size etc are not presented; the analysis is streightforward but simple; no attempts were made to corroborate the (flawed) opinion of the authors that co-infection or colonisation with these potential pathogens impair wound healing; e.g., no logistic regression models were provided to suggest that harbouring these pathogens was associated with delayed or impared wound healing.

For this reviewer, it remains unclear whether positive cultures were obtained in specific patient groups; and how many turned out to have a negative culture result. We only see the results of cultures, no breakdown with study participants as the denominator . .

Reviewer #2: Yes

Reviewer #3: Results generally well presented but there is some missing data

**Conclusions**

-Are the conclusions supported by the data presented?

-Are the limitations of analysis clearly described?

-Do the authors discuss how these data can be helpful to advance our understanding of the topic under study?

-Is public health relevance addressed?

Reviewer #1: The Conclusion itself is supported by the data provided.

In The Discussion section, many aspects are discussed that were actually not addressed by the study team; the mentioning that drug-resistant pathogens are a threat to the health care system may be true, but this seems speculative as this aspects was not covered in the analysis. Their mentioning and discussing antimicrobial stewardship is therefore a bit of a long shot . . One might argue that stricter adherence to appropriate wound care could suffice; others have demonstrated (using Whole Genome Sequencing) that in-hospital transmission occurred, at least with MRSA; the authors themselves have not provided added information to corroborate this.

Reviewer #2: Yes

Reviewer #3: Conclusions need some clarifications as described below

**Editorial and Data Presentation Modifications?**

Reviewer #1: The work represents a confirmation of earlier work by other groups that indeed, potential bacterial drug-resistant pathogens can be isolated from Buruli ulcer lesions; the data set is not extremely rich, and lacks sophisticated data analysis that would potentially support existing hypotheses, notably, the hypothesis that these organisms impair wound healing in BU. A flow chart and a per-study-participant table would help the reader to appreciate the work more meaningfully.

Reviewer #2: Minor Revision

Reviewer #3: Minor modifications:

- Line 155: Why were pregnant or breast-feeding women excluded from the study?

- Line 197: suggest change to Among these, 33 (59%) were males as no need to repeat 33/56

- Line 241: None of the increases are statistically significant (p>0.05). This should be clarified.

- Line 251: need to clarify that the stats provided here for tigecycline and nitrofurantoin are for resistance, as it reads from earlier in the sentence that they are susceptible.

- Line 262: None of the increases are statistically significant (p>0.05). This should be clarified.

- Line 275: In table 3 oxacillin sensitivities are not included, yet reported in Line 262.

- Line 292: suggest modify to “insights into the local resistance patterns associated with BU in Ghana…”

- Line 294: suggested modify to “was slightly higher than previously reported.”

- Line 295: majority of patients were not <15 years (only 39.3%)

- Line 322: suggest modify to “, along with the provision…”

- Line 340: suggest change to “were observed against tetracyclines”

- Line 343: suggest change to “After treatment (week 16)…”

**Summary and General Comments**

Reviewer #1: Strengths: sample size is larger than their earlier study, with a nice longitudinal design with sampling at three different time points;

Novelty: confirmatory, not novel;

Significance: not entirely clear, as we already know that drug-resistant bacterial isolates can be abundantly recovered from BU lesions; we also know that current practice to start simple antimicrobials for suspected secondary infection is often inappropriate and potentially harmful; we still don't know whether culture-based targeted therapy brings any good for patients, and whether wound healing would be improved; only a controlled clinical trial would be able to provide that answer, or if that is unfeasible, perhaps, a data-rich, large clinical database of considerable size (c.f., the Individual Patient Data analysis that has been used in MDRTB) could be used to conduct a sophisticated multivariable analysis to provide circumstantial evidence to support the hypothesis that such therapy might be beneficial.

Weaknesses: culture-based data are difficult to interpret; quantitative genomic analysis might be a much more appropriate study design, but obviously this might be unaffordable; sampling technique is critical, and the authors should either exactly explain the sampling technique, or admit that leaving it to the discretion of the attending study team member on the ground might introduce noise, or even bias.

Reviewer #2: Abstract

• Background: there are some redundancy with “antimicrobial resistance”

• Methodology: “ESBL production” can you provide the definition of “ESB;” as is the first apparition on the manuscript. Same for “MRSA” and “MDR”

Methods

• Improved the presentation of tables 2 and 3. reading and understanding are not always easy

Discussion

• Line 304 “Yeboah-Manu et al” must be written in italics and the same applies to the entire document

General comments:

• Quality of the images should be ameliorated

• Use the same police to presented the references

Reviewer #3: In general, an interesting paper to demonstrate the type and prevalence and the antibiotic sensitivities of bacterial colonising flora of BU wounds pre, during and post BU treatment in Ghana.

However, I think that there are some points that would be important to address as follows:

- It is important that the reader understands that the type and prevalence of organisms and their antibiotic sensitivities may vary depending on location. Thus the organisms isolated in this study are reflective of the situation in regional Ghana but not necessarily reflective of what may be found in other settings (as recognised in line 307). It should be mentioned therefore that when advocating for this information to be utilised to develop treatment guidelines, it should be recognised that to maximise usefulness, local settings will need to do their own surveillance microbiological data.

- It is regularly stated in the paper (eg lines 282, 313, 362, 367) that there was an increase in MRSA from baseline to Wk 16 – however the increase is not statistically significant and may just represent chance, especially with the small sample size (6/12 v 9/15, p-value >> 0.05). These statements need to be modified to reflect this.

- Likewise in line 220 the increase in P. mirabilis from 7/43 to 10/50 is not statistically significant (p>>0.05), and in Line 281 the decrease in ESBLs does not reach statistical significance.

- In table 3, why were only 2 isolates tested for rifampicin – it is important, especially on treatment, to understand if there is any increase in resistance in colonizing flora to the antibiotics used in treating BU (ie rifampicin and clarithromycin). This is important AMR information. Further to this it would be worth noting the results from table 3 that show no significant increase in macrolide resistance in S. aureus isolates on treatment.

- Interesting to note in Line 219 that the proportion of S. aureus increased and this is statistically significant (P<0.05) which would be worth mentioning. This is a little unexpected in view of rifampicin and clarithromycin’s usual anti-staph activity – maybe worth further explaining this fact in line 316 of discussion. However I think it is important to consider that it may also be due to the induction of resistance in these organisms (especially to rifampicin as not to clarithro according to table 3). Thus a bit more discussion about what the AMR implications for S. aureus are with current BU treatments would be important.

PLOS authors have the option to publish the peer review history of their article (what does this mean? ). If published, this will include your full peer review and any attached files.

**Do you want your identity to be public for this peer review?** For information about this choice, including consent withdrawal, please see our Privacy Policy .

Reviewer #1: No

Reviewer #2: No

Reviewer #3: No

**Figure resubmission:**
---

## [Editor Report · Decision Letter 1]

29 Apr 2025

PNTD-D-25-00354R1Antimicrobial resistant bacteria isolated from Buruli ulcer lesions in GhanaPLOS Neglected Tropical DiseasesDear Dr. Amoako, Thank you for submitting your manuscript to PLOS Neglected Tropical Diseases. After careful consideration, we feel that it has merit but does not fully meet PLOS Neglected Tropical Diseases's publication criteria as it currently stands. Therefore, we invite you to submit a revised version of the manuscript that addresses the points raised during the review process. Please submit your revised manuscript within 30 days May 29 2025 11:59PM. If you will need more time than this to complete your revisions, please reply to this message or contact the journal office at plosntds@plos.org. Please include the following items when submitting your revised manuscript: * A rebuttal letter that responds to each point raised by the editor and reviewer(s). You should upload this letter as a separate file labeled 'Response to Reviewers '. This file does not need to include responses to any formatting updates and technical items listed in the 'Journal Requirements' section below. * A marked-up copy of your manuscript that highlights changes made to the original version. You should upload this as a separate file labeled 'Revised Manuscript with Track Changes '. * An unmarked version of your revised paper without tracked changes. You should upload this as a separate file labeled 'Manuscript '. If you would like to make changes to your financial disclosure, competing interests statement, or data availability statement, please make these updates within the submission form at the time of resubmission. Guidelines for resubmitting your figure files are available below the reviewer comments at the end of this letter. We look forward to receiving your revised manuscript. Kind regards, Paul J. ConverseAcademic EditorPLOS Neglected Tropical Diseases Stuart BlacksellSection EditorPLOS Neglected Tropical Diseases

Shaden Kamhawi

co-Editor-in-Chief

Paul Brindley

co-Editor-in-Chief

**Additional Editor Comments:** Dear Amoako and colleagues,

Your revised version appears to adequately answer the reviewers' concerns. However, you did not include Figure 1. Please provide the figure to the journal, assuming that it now meets all specifications.

Thank you for submitting your work to PLoS Neglected Tropical Diseases.**Journal Requirements:**

Please upload a copy of Figure Figure 1 which you refer to in your text on page 6 and 7. Or, if the figure is no longer to be included as part of the submission please remove all reference to it within the text.

**Reviewers' comments:** **Figure resubmission:** While revising your submission, please upload your figure files to the Preflight Analysis and Conversion Engine (PACE) digital diagnostic tool, https://pacev2.apexcovantage.com/. PACE helps ensure that figures meet PLOS requirements. To use PACE, you must first register as a user. Registration is free. Then, login and navigate to the UPLOAD tab, where you will find detailed instructions on how to use the tool. If you encounter any issues or have any questions when using PACE, please email PLOS at figures@plos.org. Please note that Supporting Information files do not need this step. If there are other versions of figure files still present in your submission file inventory at resubmission, please replace them with the PACE-processed versions.**Reproducibility:** To enhance the reproducibility of your results, we recommend that authors of applicable studies deposit laboratory protocols in protocols.io, where a protocol can be assigned its own identifier (DOI) such that it can be cited independently in the future. Additionally, PLOS ONE offers an option to publish peer-reviewed clinical study protocols. Read more information on sharing protocols at https://plos.org/protocols?utm_medium=editorial-email&utm_source=authorletters&utm_campaign=protocols

---

## [Editor Report · Decision Letter 2]

13 May 2025

Dear Amoako,

We are pleased to inform you that your manuscript 'Antimicrobial resistant bacteria isolated from Buruli ulcer lesions in Ghana' has been provisionally accepted for publication in PLOS Neglected Tropical Diseases.

Best regards,

Paul J. Converse

Academic Editor

Stuart Blacksell

Section Editor

Shaden Kamhawi

co-Editor-in-Chief

Paul Brindley

co-Editor-in-Chief

Dear Dr. Amoako and colleagues,

Thank you for taking action on the last comment and for submitting your work to PLoS Neglected Tropical Diseases.

---

## [Editor Report · Acceptance letter]

Dear Amoako,

We are delighted to inform you that your manuscript, "Antimicrobial resistant bacteria isolated from Buruli ulcer lesions in Ghana," has been formally accepted for publication in PLOS Neglected Tropical Diseases.

Best regards,

Shaden Kamhawi

co-Editor-in-Chief

Paul Brindley

co-Editor-in-Chief
